# Spatial and Chronological Assessment of Variations in Carbon Stocks in Land-Based Ecosystems in Shandong Province and Prospective Predictions (1990 to 2040)

Xiaolong Xu [1,†], Kun Li [2,3,†], Chuanrong Li [2,3,*], Fang Han [1], Junxin Zhao [1] and Youheng Li [1]

[1] School of Civil Engineering and Geomatics, Shandong University of Technology, Zibo 255000, China; 23407010844@stumail.sdut.edu.cn (X.X.); hanf@lreis.ac.cn (F.H.); 23507020863@stmail.sdut.edu.cn (J.Z.); 23507020855@stumail.sdut.edu.cn (Y.L.)

[2] Taishan National Forest Ecosystem Research Station for Long-Term Observation, Taian 271018, China; kunli@sdau.edu.cn

[3] The Research Center for Forest Carbon Neutralization Engineering of Shandong Higher Education Institutions, Key Laboratory of Ecological Protection and Safety Prevention in the Lower Yellow River of Shandong Higher Education Institutions, Taian 271018, China

\* Correspondence: chrli@sdau.edu.cn

† These authors contributed equally to this work.

**Abstract:** Analyses of regional carbon stock dynamics, particularly of spatial and temporal dynamics and their relationship with land use transitions, play a key role in the management of terrestrial ecosystem functions and the optimization of land resource allocation. This study focuses on Shandong Province, an important ecological security barrier along the eastern coast of China, to explore carbon stock changes and how land use modifications contributed to the chrono-spatial distribution of carbon stocks from 1990 to 2020, with additional forecasts up to 2040. Based on Natural Variation Conditions, Ecological Variation Conditions, and the City's Variation Conditions, the results indicate a downward trend in carbon stocks across Shandong Province, from $2661.87 \times 10^6$ t in 1990 to $2380.02 \times 10^6$ t in 2020. Carbon stocks exhibit a highly uneven spatial distribution, with concentrations being notably higher in the central and eastern regions. Cities are classified based on their carbon stock level: high carbon stock cities (Linyi, Weifang, Yantai), large carbon stock cities (Jinan, Jining, Qingdao, Dezhou, Binzhou, Liaocheng, Taian, Zibo, Dongying), and cities with general carbon stock levels (Weihai, Rizhao, Zaozhuang). The major driver of carbon stock decline is the conversion of ecological lands into urban areas, with cultivated lands and forests being the primary carbon storage contributors. Projections suggest that under the City's Variation Conditions, carbon stocks will decrease from $2380.02 \times 10^6$ t in 2020 to $1654.16 \times 10^6$ t by 2040, while Carbon stocks will rise from $2380.02 \times 10^6$ t to $2430.56 \times 10^6$ t under the Ecological Variation Conditions. A significant disparity in carbon sink potential is found across cities, which are divided into high carbon sink potential cities (Yantai, Dezhou, Weifang, Qingdao, Jinan), large carbon sink potential cities (Binzhou, Weihai, Zibo, Liaocheng, Dongying, Linyi, Taian, Rizhao, Zaozhuang), and general potential cities (Jining, Heze). The insights gained from this study are essential for promoting the conservation of regional terrestrial ecosystems, directing land use policy development, and supporting sustainable development initiatives in Shandong Province.

**Keywords:** land use; carbon stocks; spatial and temporal variability; multi-circumstances forecasting; natural ecosystem

# 1. Introduction

Carbon accumulation in earth's terrestrial environments is pivotal to the global carbon cycle, acting as an essential indicator for assessing regional ecosystem services [1]. Land use pattern changes are significant drivers of the variability in carbon stocks across both space and time. Changes in land use patterns directly influence vegetation types and their spatial distributions, thereby affecting carbon stock levels [2,3]. Therefore, precisely assessing the effects of land utilization variations on regional carbon stocks is crucial for optimizing land resource management and achieving dual carbon goals. A comprehensive analysis of the connection between carbon stocks and shifting land utilization can offer valuable insights for sustainable development.

Traditional methods for studying carbon stock changes, such as sample inventories, eddy covariance, and box models, while precise, often fall short in terms of capturing the dynamic responses of carbon stocks to land use changes over broad spatial and temporal scales [4,5]. Consequently, model simulation techniques have gained popularity for estimating regional carbon stocks due to their flexibility and suitability across various spatial extents [6]. Examples of widely used land utilization simulation models are CLUE-S, FLUS, and SD-CLUS-S [7–9]. Due to the dynamic variability of the landscape in Shandong Province, these models often struggle to illustrate the complex interactions among the various land utilization categories [10]. In contrast, the CA-Markov model effectively integrates the spatial simulation strengths of Cellular Automata with the long-range predictive power of the Markov process, thereby improving prediction accuracy and addressing these challenges. The InVEST model, recognized for its ability to quantify the interconnection between carbon stocks and changes in land utilization, is especially relevant to Shandong Province due to the region's diverse ecosystems and varied land use patterns [11]. This model has been widely adopted for carbon stock estimations in diverse regions due to its user-friendly interface and robust visualization capabilities [6,7]. Recent studies have successfully integrated the InVEST model with the CA-Markov model to assess changes in carbon stocks across space and time and predict future land use scenarios across various regions in China [12–14]. These combined methodologies offer high accuracy and efficiency, facilitating large-scale monitoring of carbon stock fluctuations and enabling comparative analyses over extended periods, thus providing a solid foundation for future land planning [15,16].

Currently, many regions in China have carried out studies based on changes in carbon stocks based on land use types [17–24]. However, there are large differences between regions; for example, the reduction of arable land in some regions may lead to an increase in carbon stocks, while in others it may lead to a decrease. Therefore, we found that there is significant spatial heterogeneity in carbon stock changes based on land utilization types. Shandong Province, located on the eastern coast of China, is a major agricultural and industrial hub, marked by a high population density and rapid urbanization. Previous research has generally overlooked comprehensive analyses of carbon stocks across entire terrestrial ecosystems in Shandong Province and long-term trends in these changes [25–29]. Analyzing carbon stock changes in Shandong's terrestrial ecosystems can provide valuable insights into the spatial effects of urban growth and eco-protection on carbon stocks, which can support the achievement of carbon neutrality within the framework of national ecological security [30]. Therefore, the aim of this study is to capitalize on land utilization data collected between 1990 and 2020—combining the CA-Markov and InVEST models to analyze the spatial and temporal variability of carbon stocks in Shandong Province and the response relationship between land utilization variations and carbon stocks—to explore the differences in the impact of land use types on carbon stocks among regions and to summarize the characteristics of the significance of land utilization types on carbon

stocks in the study area. The study also seeks to forecast and model land use patterns and changes in carbon stocks for the years 2030 and 2040 across different scenarios, thus providing a detailed analysis of carbon stock variations. This research is expected to offer important insights for ecological development and dual-carbon strategy planning in Shandong Province.

## 2. Materials and Methods

### 2.1. Researh Region

Shandong Province (34°23′38°24′ N, 114°48′122°42′ E) is situated on the eastern coast of China, at the confluence of the Haihe River, Yellow River, and Huaihe River basins. The province features varied topography, exhibiting elevated terrain in the southwest and lower altitudes to the northeast. It experiences a warm, temperate monsoon climate, where perennial temperatures vary between 11 °C and 14 °C and perennial precipitation varies from 550 mm to 950 mm. These favorable natural conditions, combined with its strategic geographical location, make Shandong a key region for biodiversity conservation. As a critical component of China's national ecological security strategy, Shandong serves vital functions as part of the eastern ecological security barrier and the ecological protection zone at the Yellow River estuary. This role is essential for the development of ecological corridors across the coastal zone, the Yellow River basin, and the Lunan mountainous region. These efforts are crucial for ensuring ecological security in the eastern region and contribute significantly to the overall stability and sustainable development of national ecological security [31] (Figure 1).

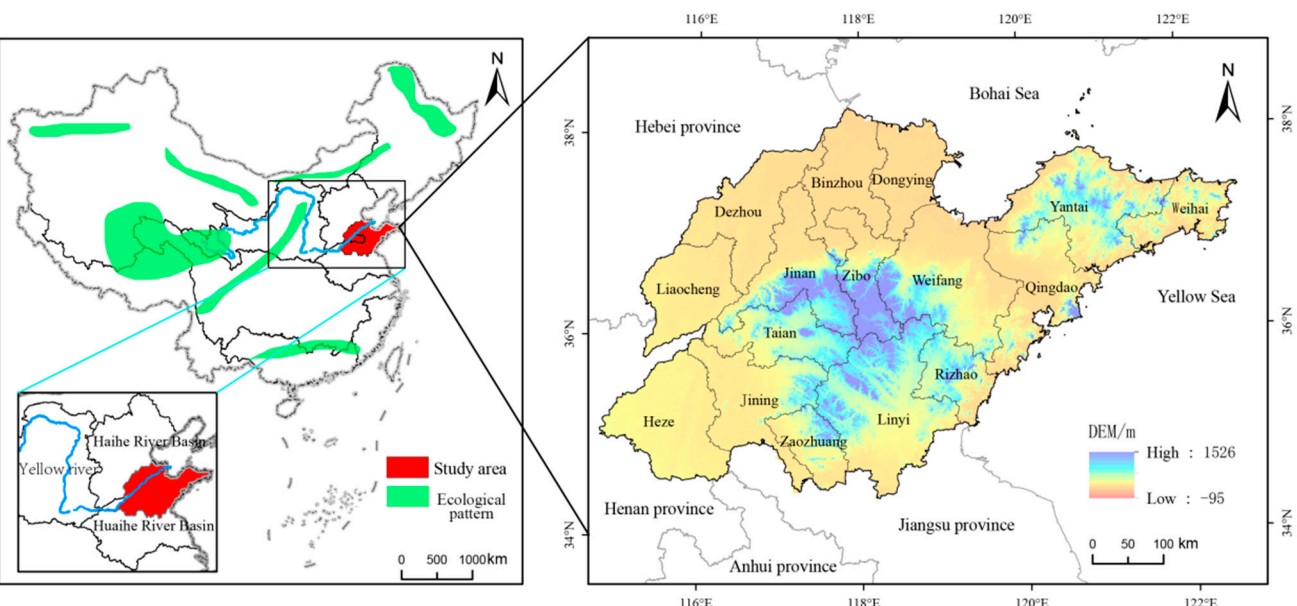

**Figure 1.** Schematic representation of the research locale.

### 2.2. Research Methods

2.2.1. Data Preprocessing

(1) The temperature and precipitation data were transformed from vector points to raster format by kriging interpolation, with a geospatial resolution of 1 km. Some of the missing null points were supplemented using mean interpolation and identifying and removing outliers using IRQ Methods. This transformation facilitated the calculation of the average annual temperature (7.56 °C for Shandong Province and 7.10 °C for the national average) and annual precipitation (678.34 mm for Shandong Province and

456.26 mm for the national average) for the period from 1990 to 2020. These data were then used to adjust the carbon density coefficients.

(2) Population density and GDP data were similarly (with temperature and precipitation) transformed from vector points to raster format, with a consistent geospatial resolution of 1 km.

(3) Elevation and slope data were adjusted through resampling to conform to the 30 m geospatial resolution of the land use classification.

(4) Information on the highway network, railway network, and river systems was extracted in shp format and cropped into the administrative boundaries of Shandong Province.

All the above data and land use data were resampled in ArcGIS to a uniform resolution of 1 km and a consistent spatial extent for proper analysis in the IDRISI 17.2 software (Table 1).

**Table 1.** Data message.

| Datatype | Data Attributes | Period | Geospatial Resolution | Data Origin |
|---|---|---|---|---|
| Land use type | — | 1990–2000 2010–2020 | 30 m | http//www.resdc.cn |
| Climatic data | Temperature | 1990–2020 | 1 km | http://www.geodata.cn |
| | Precipitation | 1990–2020 | 1 km | |
| Topography | Elevation | — | 90 m | http://www.gscloud.cn |
| | Slope | — | 90 m | |
| Socio-economic | Population density | 1990–2020 | 1 km | http://www.geodata.cn |
| | GDP | 1990–2020 | 1 km | |
| | Highway | 2020 | — | |
| | Railroad | 2020 | — | |
| | River | 2020 | — | |
| carbon density | — | — | — | Table 2 |

**Table 2.** Carbon density values (t/hm$^2$) [25–29,32–34].

| Land Use Type | $C_{above}$ | $C_{below}$ | $C_{soil}$ | $C_{dead}$ |
|---|---|---|---|---|
| Cultivated land | 17.00 | 80.72 | 108.42 | 9.82 |
| Forest | 42.44 | 115.92 | 158.82 | 14.11 |
| Grassland | 35.30 | 86.53 | 99.90 | 7.28 |
| Waters | 0.30 | 0.00 | 0.00 | 0.00 |
| Construction land | 2.51 | 27.50 | 0.00 | 0.00 |
| Unused land | 1.30 | 0.00 | 21.61 | 0.00 |

2.2.2. Invest Model

The InVEST model is a comprehensive framework for evaluating habitat quality, encompassing several interrelated modules, including carbon stock assessment, habitat quality analysis, and soil and water conservation [35]. In this study, we focused on the carbon stock module to analyze variations in carbon stocks within Shandong Province's terrestrial ecosystems over a thirty-year period (1990–2020). This block categorizes ecosystem carbon stocks into four sections: $C_{above}$, $C_{below}$, $C_{soil}$, and $C_{dead}$ [36,37].

The modeling procedure begins with the establishment of the carbon stock calculation formula, as shown in Equations (1) and (2):

$$C = C_{above} + C_{below} + C_{soil} + C_{dead} \tag{1}$$

$$C = (C_{abovei} + C_{belowi} + C_{soili} + C_{deadi}) \times S_i \tag{2}$$

In this equation, *i* represents the average carbon intensity associated with each utilization patterns of land, and $S_i$ denotes the region corresponding to that utilization patterns of land. The carbon stock module requires the integration of land management classifications and carbon concentrations into the software for computational purposes. The carbon density values vary due to differences in geological environments and climatic conditions across regions [38]. Therefore, it is essential to adjust the carbon density coefficients based on the climatic characteristics of Shandong Province, in accordance with the findings of other scholars [39–41].

$$C_{sp} = 3.3968 \times P + 3996.1 \tag{3}$$

$$C_{bp} = 6.7981e^{0.00541P} \tag{4}$$

$$C_{bt} = 28 \times T + 398 \tag{5}$$

where $C_{sp}$ represents the soil carbon density (kg·m$^{-2}$), while $C_{bp}$ and $C_{bt}$ represent biomass carbon densities (kg·m$^{-2}$). Here, *P* denotes the perennial mean precipitation (mm), and *T* refers to the perennial mean temperature (°C):

$$K_{bp} = \frac{C'_{bp}}{C''_{bp}} \tag{6}$$

$$K_{bt} = \frac{C'_{bt}}{C''_{bt}} \tag{7}$$

$$K_b = K_{bt} \times K_{bp} \tag{8}$$

$$K_s = \frac{C'_{sp}}{C''_{sp}} \tag{9}$$

In this context, $K_{bp}$ and $K_{bt}$ are the calibration factors for biomass and soil carbon density. The variables $C'_{bp}$, $C''_{bp}$, $C'_{bt}$, and $C''_{bt}$ represent biomass carbon density data derived from perennial precipitation and temperature data for both Shandong Province and the national average. Additionally, $C'_{sp}$, $C''_{sp}$ represent soil carbon density data for both Shandong Province and the entire country. The coefficients $K_b$ and $K_s$ are the adjustment factors for biomass and soil carbon densities in Shandong Province relative to national averages. The results after corrections are presented in Table 3.

**Table 3.** Corrected Carbon Density Values (t/hm$^2$).

| Land Use Type | $C_{above}$ | $C_{below}$ | $C_{soil}$ | $C_{dead}$ |
|---|---|---|---|---|
| Cultivated land | 13.77 | 65.37 | 103.40 | 7.95 |
| Forest | 34.34 | 93.88 | 151.49 | 11.43 |
| Grassland | 28.59 | 70.06 | 95.30 | 5.90 |
| Waters | 0.24 | 0.00 | 0.00 | 0.00 |
| Construction land | 2.02 | 22.27 | 0.00 | 0.00 |
| Unused land | 1.05 | 0.00 | 20.61 | 0.00 |

The sixteen cities in Shandong Province were ranked in descending order based on their total carbon stock and categorized into three groups: high carbon stock cities (>2 × 10$^8$ t), large carbon stock cities (1 × 10$^8$ t to 2 × 10$^8$ t), and general carbon stock cities (<1 × 10$^8$ t), using the standard deviation method.

### 2.2.3. CA-Markov Modeling

(1)  Principle of the Model

The CA-Markov model makes use of a transition matrix to represent changes in land utilization over time, effectively combining the cartographic simulation strengths of the Cellular Automata model with the temporal forecasting capabilities of the Markov model. This integration boosts the model's effectiveness in terms of predicting and simulating the dynamics of land use transformations across different spatial and temporal contexts [14]. The CA model excels at illustrating the dynamic processes over the spatial and temporal dimensions of various natural processes, including land use [42]. The model can be mathematically defined as stated below:

$$U(x, x+1) = f[U(x), T]  \tag{10}$$

where $U$ refers to the limited set of discrete states available to the cell, $f$ defines the transition rule governing the behavior of the cell states, $T$ refers to the collection of adjacent cells surrounding each cell, and $x$, $x+1$ refer to two distinct time periods. Markov chains are grounded in the Markov stochastic process, which serves as the primary tool for predicting landscape changes [43]. The related calculation formulas are:

$$P_{ij} = \begin{bmatrix} P_{11} & P_{12} & \cdots & P_{1n} \\ P_{21} & P_{21} & \cdots & P_{2n} \\ \cdots & \cdots & \cdots & \cdots \\ P_{n1} & P_{n2} & \cdots & P_{nn} \end{bmatrix} \text{ and } \sum_{j=1}^{n} P_{ij} = 1 (i, j = 1, 2, \cdots, n)  \tag{11}$$

$$S_{x+1} = P_{ij} \times S_x  \tag{12}$$

where $S_x$ and $S_{x+1}$ denote the land utilization conditions at periods $x$ and $x+1$, respectively, $P_{ij}$ is the matrix representing transfer probabilities, and $n$ represents the land use classification.

(2)  Accuracy verification

The Kappa index was employed to assess the precision of the simulation data. This assessment was conducted by inputting actual land use classifications for 1990, 1995, 2000, 2005, 2010, 2015 and 2020 into the Crosstab module of IDRISI 17.2 software and comparing the results with the predicted outcomes. The Kappa coefficients under the Natural Variety Circumstances (NVC) scenario were 0.8912, 0.9242, 0.9172, 0.9065, 0.9261, and 0.9329, indicating strong simulation performance.

(3)  CA-Markov model scenario setting

In alignment with the status of the urbanization process, ecological protection policies, and empirical methods from related studies [13,44], three land use change scenarios were established for Shandong Province:

Scenario 1: Natural Variety Circumstances (NVC). This scenario assumes that the factors influencing land use change from 2020 to 2040 will remain relatively stable. It predicts land use changes for 2030 and 2040 based upon a probability matrix derived from land use shifts between 2010 and 2020.

Scenario 2: Ecological Variety Circumstances (EVC). This scenario simulates the utilization patterns of land changes in Shandong Province for 2030 and 2040 under ecological conservation policies that limit the switch of forest and grassland to other categories, while permitting other categories to be transformed into ecologically significant land.

Scenario 3: City's Variety Circumstances (CVC). This scenario emphasizes land use changes driven by urban expansion resulting from urban planning and development in

extreme circumstances. It predicts that by 2030 and 2040, all land classes, except water bodies, will be converted into construction land.

The differences in carbon stock projections for the sixteen cities in Shandong Province under the EVC the CVC for 2040 were ranked in descending order and classified into three categories: cities with high carbon sink potential, cities with moderate carbon sink potential (0 t to $5 \times 10^7$ t), and cities with low carbon sink potential (<0 t). These categories were determined using the standard deviation method.

2.2.4. Carbon Stock Dynamic Change Rate

The rate of dynamic change in carbon stocks was used to describe the variation in carbon stocks over a specified time periods, effectively reflecting the intensity of spatial dynamic changes in carbon stocks [45]. A one-dimensional linear regression model could be applied to analyze the linear connection between the variable and time [46]. The integration of these two methods supplies a solid foundation for understanding the geographical and chronological trends of variations in carbon stocks. The relevant formulas are as follows:

$$R = \frac{U_b - U_a}{U_a} \times \frac{1}{T} \times 100\% \tag{13}$$

$$\theta_{slope} = \frac{n \times \sum\limits_{i=1}^{n} i \times R_i - \sum\limits_{i=1}^{n} i \sum\limits_{i=1}^{n} R_i}{n \times \sum\limits_{i=1}^{n} i^2 - (\sum\limits_{i=1}^{n} i)^2} \tag{14}$$

where $R$ represents the yearly rate of carbon stock variation within the study timeframe; $U_a$ and $U_b$ denote the carbon stocks at the outset and the close of study timeframe; $T$ denotes study timeframe; $\theta_{slope}$ represents the slope of the dynamic change in carbon stocks; $n$ is the length of the temporal data; and $R_i$ signifies the rate of variation in carbon stock dynamics during year $i$. A positive slope indicates an increasing trend in the dynamic change of carbon stocks over time, while a negative slope reflects a decreasing trend.

## 3. Results

*3.1. Examining the Traits of Land Utilization Category Changes in Shandong Province over the Period from 1990 to 2020*

The dominant land utilization categories in Shandong Province are cultivated land, construction land, forest, and grassland. From 1990 to 2020, crucial alterations occurred in the areas of each land utilization category. Specifically, cultivated land decreased by 16,089.68 km², while grassland and unused land declined by 2609.31 km² and 1557.99 km², respectively. In contrast, forests and building sites increased by 1373.98 km² and 16,507.78 km², respectively, and water land expanded by 2375.21 km². An analysis of the land utilization transition matrix showed that the increase in construction land (17,895.20 km²) during the study timeframe was primarily due to the conversion of cultivated land. Additionally, the growth of forested regions (4112.71 km² and 762.06 km²) was primarily attributed to the transformation of cropland and grassland, while unused land along the northern coast was predominantly transformed into water bodies (764.13 km²) (see Figure 2).

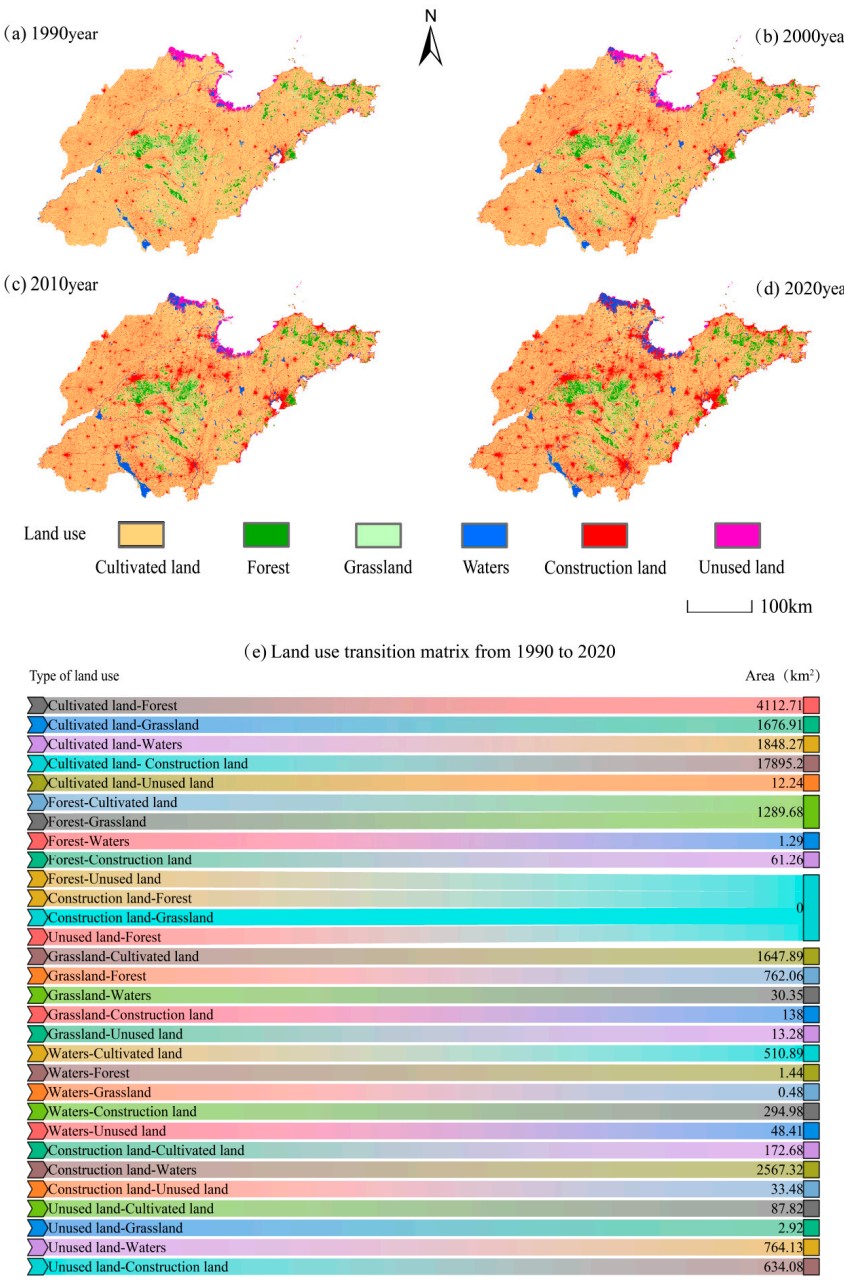

**Figure 2.** Changes in land use in Shandong Province, 1990–2020.

## 3.2. Characteristics of Location-Based and Time-Related Variations in Carbon Stocks in Terrestrial Ecosystems in Shandong Province from 1990 to 2020

### 3.2.1. Characteristics of Temporal Changes in Carbon Stock in Shandong Province

During the period from 1990 to 2020, the cumulative carbon reserve of terrestrial ecosystems in Shandong Province declined by $281.86 \times 10^6$ t, reflecting an overall decrease of 0.11%. From 2000 to 2010, the carbon stock decreased from $2572.05 \times 10^6$ t to $2458.13 \times 10^6$ t, marking the most significant decline (0.04%). This decline was attributed to several interrelated factors. First, the enlargement of construction land gave rise to a direct loss of carbon stocks (−15,810.23 t) due to the conversion of carbon-rich ecosystems such as forests and grasslands into urban areas. Second, the decrease in cultivated land per unit area [31], largely driven by urbanization and agricultural land abandonment, contributed to a loss of 9896.51 t in carbon stocks. Additionally, the increase in forest carbon stocks during this period (17,585.78 t), although significant, was not sufficient to offset the losses from other land-type changes. The overall decline also reflected broader land-use changes

and may have been influenced by regional climate patterns, such as reduced precipitation and temperature fluctuations, which can directly affect soil carbon storage and vegetation growth [28].

Regarding the distribution of carbon reserves across different land utilization categories, cultivated land consistently accounted for the largest proportion throughout the 30-year period, ranging from 87.59% to 85.09%, followed by forest (7.05% to 9.56%), grassland (3.56% to 1.79%), and construction land (1.61% to 3.45%). The proportions of waters and unused area were relatively low, i.e., not exceeding 1%. The dynamic rate of change in carbon stocks, in descending order, was as follows: cultivated land (−38.38%), grassland (−6.53%), construction land (5.02%), forest (5.10%), unused land (−0.42%), and water bodies (0.01%). Notably, the transformation of carbon stocks in cultivated land were significantly more pronounced than those in other land types.

### 3.2.2. Characteristics of Spatial Variations in Carbon Reserves in Shandong Province

From 1990 to 2020, while the overall geographical distribution pattern of carbon stock in Shandong Province remained relatively stable, significant local variations were observed (Figure 3). The central and eastern regions exhibited higher carbon stocks than other areas. Specifically, the carbon stock per unit area in the central and eastern regions increased substantially, from 13,583.32 t/km$^2$ to 23,582.30 t/km$^2$, while carbon stocks in grassland, urban fringe areas, and unused land along the eastern coastline experienced notable declines, from 7484.69 t/km$^2$ to 1940.47 t/km$^2$ (Figures 2 and 4). High-value areas of carbon stock per unit area (>16,097.65 t/km$^2$) were primarily concentrated in the forested and grassland regions of central Lu (Figure 3d) and the coastal areas of eastern Lu (Figure 3c). In contrast, low-value areas (<195.99 t/km$^2$) were found in waters, construction areas, and unused areas, particularly along the northern coastline of Laizhou Bay (Figure 3a), in the waters of Weishan Lake in the southern part of the city (Figure 3b), and in the fragmented inland distribution of construction land.

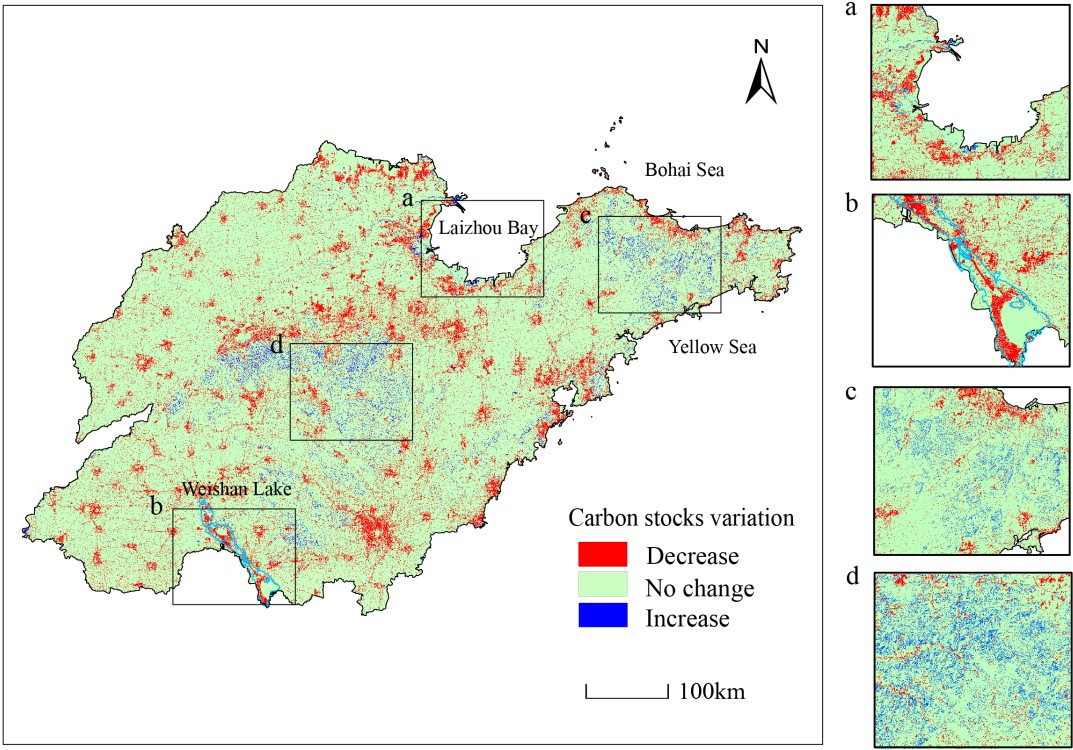

**Figure 3.** Changes in the distribution patterns of carbon stocks in Shandong Province from 1990 to 2020. (**a**). Laizhou Bay. (**b**). Weishan Lake. (**c**). Bohai Sea. (**d**). Luzhong Area.

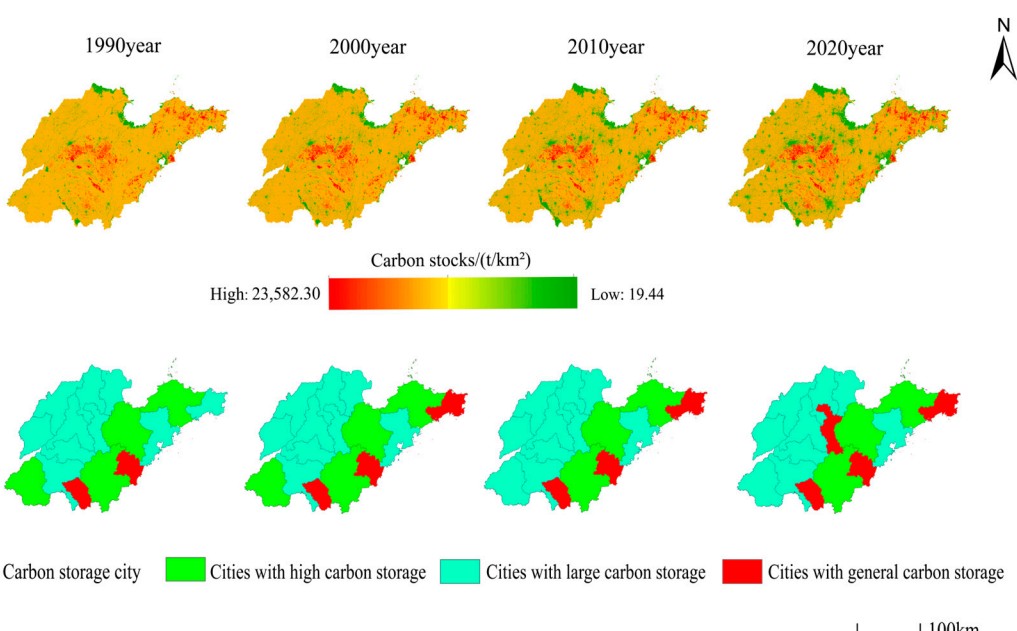

**Figure 4.** Distribution pattern of carbon stocks in land ecosystems at both grid and municipal scales in Shandong Province.

Carbon stocks across the sixteen cities in Shandong Province demonstrated a notable declining trend throughout the study period (Figure 5). Weifang City experienced the largest decrease in carbon stocks, amounting to $-33.27 \times 10^6$ t. This substantial decline can be ascribed to rapid urbanization and the metamorphosis of agricultural land into urban areas, which has led to substantial carbon loss. In contrast, Zaozhuang City recorded the smallest decline of $-7.74 \times 10^6$ t, suggesting a more stable land use pattern and effective land management practices that have helped preserve its carbon stocks. Notably, Heze City shifted from a high carbon stock category ($206.34 \times 10^6$ t) to a lower carbon stock category ($184.85 \times 10^6$ t). This transition may have been influenced by changes in land use, such as increased agricultural activities or urban expansion, which can negatively impact carbon sequestration. Similarly, Zibo City transitioned from the large carbon stock category ($110.43 \times 10^6$ t) to the average carbon stock category ($99.93 \times 10^6$ t), indicating a gradual degradation of its carbon reserves, potentially due to industrial activities and urban development [11] (Figure 4).

The contribution of cities with high carbon stocks to the overall provincial average accounted for 38.61%, with Linyi City standing out due to its impressive average annual carbon stock of $289.16 \times 10^6$ t, despite a decrease of $33.12 \times 10^6$ t over the 30-year period. This decrease could be ascribed to both natural factors such as climate variability and human factors, including agricultural land conversion and industrial expansion. Linyi's decrease rate (k = $-3.16$) indicated a relatively moderate decline compared to other cities [47]. Weifang City, exhibiting the highest decrease rate (k = $-11.25$), experienced the most significant reduction in carbon stock, totaling $33.27 \times 10^6$ t. This dramatic decline could be attributed to rapid urbanization and the conversion of carbon-rich land into urban infrastructure. Additionally, Weifang has been undergoing significant industrial development, which may have led to the degradation of local carbon sinks. In contrast, cities categorized with larger carbon stocks contributed 43.11% to the provincial average, showing that they managed to maintain relatively stable carbon reserves despite slight decreases. On the other hand, cities with general carbon stocks showed relatively smaller decreases in carbon stock (k values ranging from $-5.83$ to $-2.65$), indicating that changes in their carbon stock were comparatively stable. These cities tend to have a more balanced approach to land

use and urbanization, which may have helped preserve their carbon reserves. The more stable trends in these cities could be reflective of less rapid industrialization, better land management practices, and a more stable climate compared to other regions in Shandong Province [11] (Figure 5).

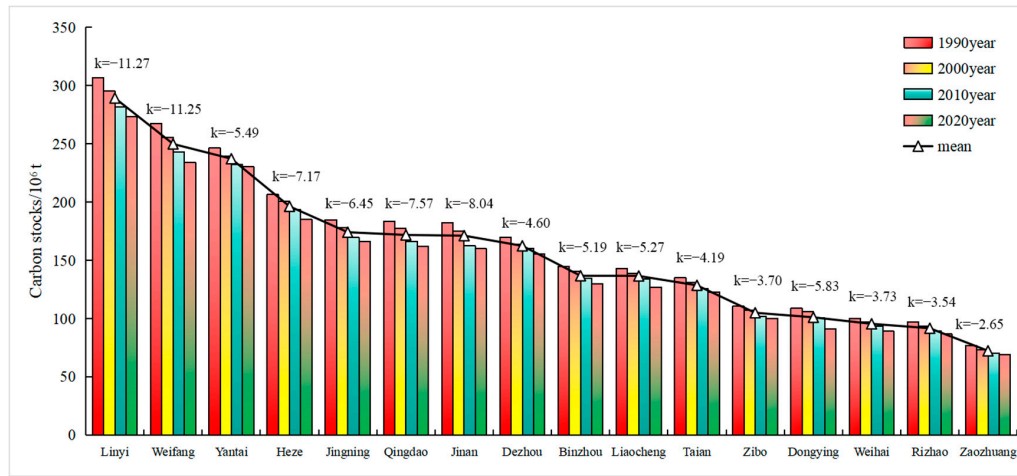

**Figure 5.** Alterations in Carbon Stocks by Municipalities in Shandong Province, 1990–2020. Note: k is the rate of change in carbon stocks from 1990 to 2020.

### 3.3. Analysis of Land Utilization Category and Carbon Stock Projections in Shandong Province Under Multiple Circumstances

3.3.1. Natural Variety Circumstances

The prediction results indicate a significant decrease within the scope of arable land under the NVC from 2020 to 2040, totaling 37,018.71 km$^2$. This reduction was primarily due to conversions to construction land and forest areas, with reductions of 34,643.11 km$^2$ and 2825.32 km$^2$, respectively. Construction land is expected to expand by 22,171 km$^2$, predominantly at the expense of arable land, which will decrease by 31,628.44 km$^2$. Additionally, forested areas are expected to expand by 4528.17 km$^2$, mainly derived from cultivated land and grassland, with respective reductions of 3993.49 km$^2$ and 1399.58 km$^2$ (Figures 6 and 7a).

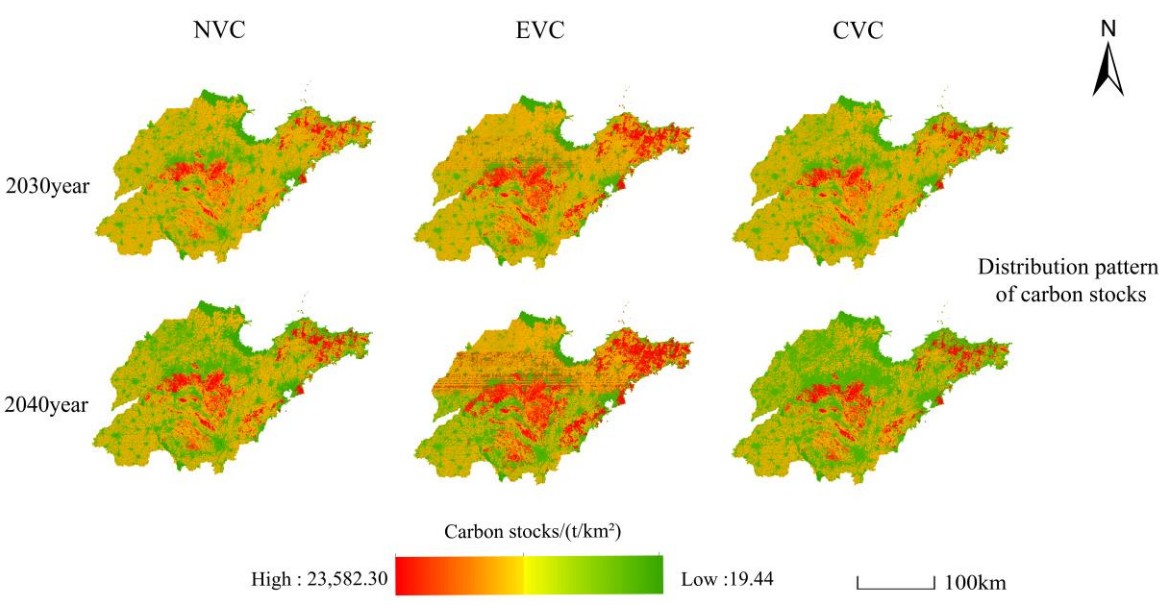

**Figure 6.** *Cont.*

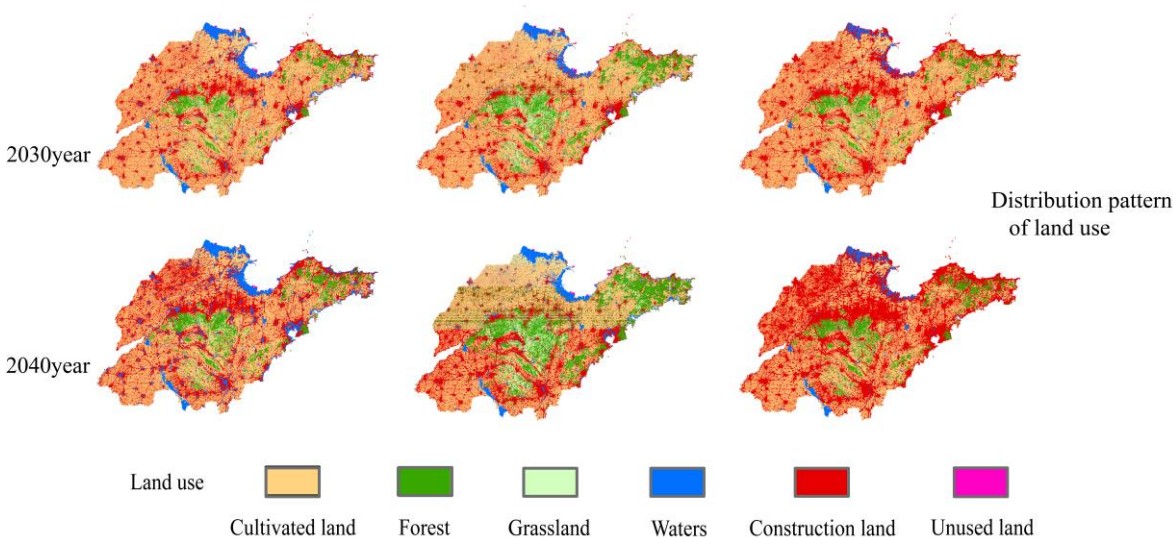

**Figure 6.** Future distribution pattern of carbon stocks and land use types in Shandong Province under multiple circumstances.

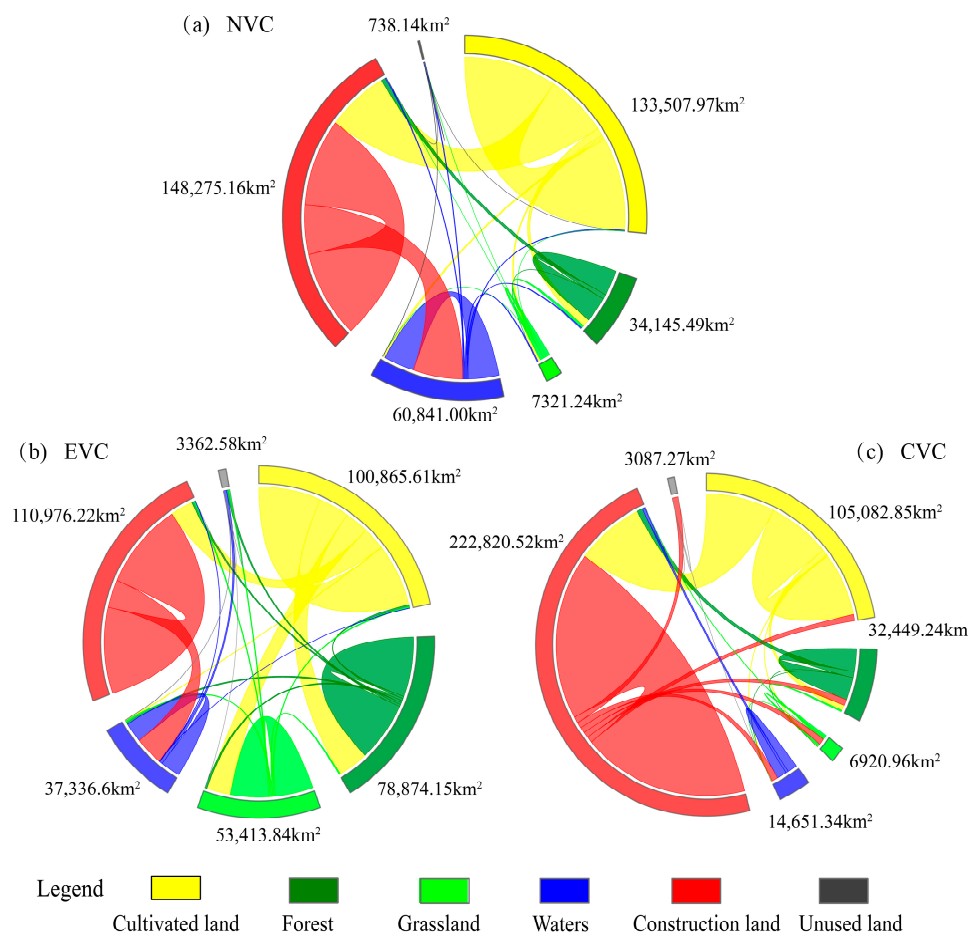

**Figure 7.** Multiple-circumstances land use transfer map for Shandong Province, 2020–2040.

### 3.3.2. Ecological Variety Circumstances

From 2020 to 2040, the carbon stock is expected to increase by $558.32 \times 10^6$ t compared to the NVC. Notably, the carbon stocks in forests and grasslands are projected to rise significantly by $497.13 \times 10^6$ t and $320.28 \times 10^6$ t, respectively. The carbon stock per unit area in the mountainous regions of central Shandong and the eastern coastal areas is also

anticipated to experience significant growth (from 13,583.32 t/km$^2$ to 23,582.30 t/km$^2$) (Figure 6). The expansion of construction land is forecast to decrease by 11,128.54 km$^2$, while cultivated areas will predominantly be turned into forested areas, grasslands, and construction land, with reductions of 14,903.20 km$^2$, 11,754.36 km$^2$, and 11,048.89 km$^2$, respectively. The proportion of ecological land is expected to increase compared to the NVC, rising from 54.16% to 64.83%, with significant expansions in forest (19,886.06 km$^2$) and grassland (15,657.54 km$^2$) areas (Figure 7b).

### 3.3.3. City's Variety Circumstances

Under the CVC, carbon stocks are projected to decrease by 80.90 × 10$^6$ t by 2030 and by 218.07 × 106 t by 2040, compared to the NVC. The ongoing outward expansion of construction land, encroaching upon existing land types, is expected to contribute to reductions in carbon stocks by 25.62 × 10$^6$ t and 58.59 × 10$^6$ t, respectively. By 2040, the rapid expansion of construction land is anticipated to further encroach on cultivated land and forested areas, resulting in a loss of 34,643.11 km$^2$ and 2862.99 km$^2$, respectively, leading to a total carbon stock loss of 254.90 × 10$^6$ t (Figures 6 and 7c).

### 3.4. Analysis of Geographical and Time-Related Variation of Carbon Stock in Shandong Province
3.4.1. Cities with High Carbon Sink Potentials

Cities with high carbon sink potential exhibited a slower decrease in carbon stocks under both the NVC and CVC scenarios. Specifically, the rates of change ranged from −2.31% to −1.39% and −2.51% to −1.54%, respectively. In contrast, a notable increase in carbon stocks was observed under the EVC, with rates of change ranging from 0.08% to 0.80%. Weifang City demonstrates relatively stable carbon stock changes across the future scenarios, owing to its substantial carbon stock base. Both Jinan and Qingdao cities transition from being characterized by larger carbon stocks to high carbon stock cities under the EVC scenario, exhibiting positive dynamic changes in carbon stocks ($\theta_{slope} > 0$). Conversely, Yantai and Dezhou cities experience significant decreases in carbon stocks under the CVC, with reductions of 78.13 × 10$^6$ t and 70.80 × 10$^6$ t, respectively, indicating sharp negative dynamic changes (Figure 8).

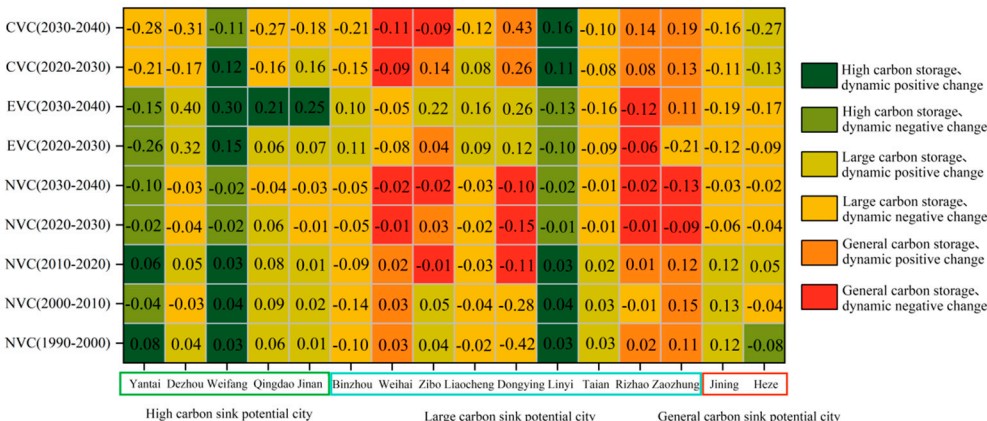

**Figure 8.** Divergence of Carbon Stocks in Shandong Province under Multiple Circumstances. Note: The value $\theta_{slope}$ represents the rate of change in carbon stock dynamics.

### 3.4.2. Cities with Larger Carbon Sink Potentials

Linyi City consistently maintained a high level of carbon stocks over the study period. In contrast, Binzhou City and Liaocheng City exhibited tendencies to stabilize their carbon sequestration capacity, with $\theta_{slope}$ values ranging from −0.11 to 0.16. Under the EVC scenario, however, both cities experience significant increases in carbon stock, amounting

to $42.38 \times 10^6$ t and $12.33 \times 10^6$ t, respectively. Taian City shows a decrease in carbon stocks across all scenarios, though the rate of dynamic change is minimal, with changes tending to stabilize. The spatial changes in carbon stocks for Rizhao and Zaozhuang are relatively consistent, but both cities see an acceleration in carbon stock reduction under the CVC scenario, with $\theta_{slope}$ changing from 0.08 to 0.14 and from 0.13 to 0.19, respectively (Figure 8).

3.4.3. Cities with General Sink Potentials

Heze and Jining cities both continue to experience a decline in carbon stocks under all three scenarios, with rates of change ranging from $-24.75\%$ to $-19.66\%$ for Heze, and $-23.68\%$ to $-21.37\%$ for Jining. Heze City has transitioned from a city with high carbon stocks to one with larger carbon stock potential. Notably, under the Ecological Variety Circumstances (EVC), the carbon stocks of Jining and Heze exhibit significant negative changes, with $\theta_{slope}$ values changing from $-0.12$ to $-0.19$ and from $-0.09$ to $-0.17$, respectively. These changes correspond to decreases of $45.75 \times 10^6$ t and $35.80 \times 10^6$ t, respectively (Figure 8).

# 4. Discussion

*4.1. Response Mechanisms of Carbon Stocks and Land Utilization Category Change*

From 1990 to 2020, land use variations in Shandong Province have had huge impacts on carbon stocks. The increase in construction land and forest areas led to a reduction of $303.91 \times 10^5$ t and an increase of $59.33 \times 10^5$ t in carbon stocks, respectively. Conversely, the decrease in cropland and grassland resulted in a reduction of $197.16 \times 10^5$ t and an increase of $7.94 \times 10^5$ t in carbon stocks, respectively. When examining the impact on a per-unit-area basis, the increase in construction land led to a decrease of 15,810.23 tons per unit area, while forest areas showed a positive increase in carbon stocks per unit area of 17,585.78 tons. Specifically, the increase in construction land, which typically involves the conversion of natural landscapes into urban or industrial areas, has resulted in a net carbon loss. This is primarily due to the metamorphosis of land for construction purposes, which results in the loss of vegetation and soil carbon stocks. Moreover, the serving of the soil surface and the reduction of plant cover severely limit the potential for carbon sequestration in these areas [44].

On the other hand, the increase in forest areas has contributed to higher carbon sequestration due to the enhanced vegetation biomass and greater soil organic carbon accumulation associated with forest ecosystems. Forest soils, rich in organic matter, typically store more carbon compared to other land use types, as the slower decomposition rates in forest environments help preserve organic carbon over time [48]. The reduction of cultivated land has given rise to a net enhancement in carbon stocks due to the cessation of soil degradation and the restoration of soil organic carbon through reduced tillage practices. This transition from cultivated land to other land uses allows soil carbon sequestration to resume, which is crucial for mitigating carbon emissions. Additionally, the decrease in grassland area resulted in a slight increase in carbon stock, as grasslands, which typically sink carbon due to their deep root systems, may have experienced carbon loss due to degradation or overgrazing [49].

Comparative studies in other provinces highlight both similar and different trends (Table 4). The conversion of other land types to construction site is the main reason for the decrease in carbon stocks, while the development and utilization of unutilized land is the main reason for the increase in carbon stocks. These regional differences highlight the role of carbon density as a primary factor influencing variations in carbon stocks. The carbon density in Northwest China is generally lower than in East China due to variations in geographical, physical, and climatic conditions [50]. Specifically,

precipitation and temperature differences between Gansu, Inner Mongolia Autonomous Region, and Shandong, Anhui lead to distinct correction coefficients for carbon density. However, these regional differences are not just a result of climatic factors, as differences in urbanization have a strong impact on land utilization conversion and carbon intensity. In Shandong Province, where the urbanization level increased from 27.30% to 63.05% in 30 years, cultivated land was primarily converted into building land (transfer area of 17,895.2 km$^2$, carbon density from 190.49 t/hm$^2$ to 24.29 t/hm$^2$), leading to a decrease in carbon stocks. The main reasons for the decline in carbon stocks in Jiangsu, Hebei, Anhui, and Hangzhou are consistent with those of Shandong province, which may be explained by the consistency of land-use change patterns between these areas and Shandong province, the small climatic differences, and the accelerated urbanization process.

**Table 4.** Statistics for selected study areas (regional carbon density in the corresponding reference) [17–24].

| Region | Year | Carbon Stock Changes | Climatic Region | Urbanization Level | Forest Cover | Main Causes of Carbon Stock Changes |
|---|---|---|---|---|---|---|
| Hangzhou City | 2000–2020 | ↓2.41 × 10$^6$ t | temperate monsoon climate | 61.17%→83.30% | 54.41%→59.43% | cropland→building land, forest→other |
| Gansu Province | 1990–2015 | ↑2.51 × 10$^6$ t | temperate continental climate | 23.54%→43.19% | 6.66%→11.33% | unused land→other |
| Inner Mongolia Autonomous Region | 2000–2020 | ↓1.01 × 10$^8$ t | temperate continental climate | 42.68%→67.48% | 17.7%→22.10% | cropland→building land |
| Fujian Province | 2000–2020 | ↓4.47 × 10$^6$ t | subtropical monsoon climate | 41.99%→68.75% | 62.96%→66.80% | cropland and grassland→ building land |
| Kunming City | 2000–2020 | ↓9.85 × 10$^5$ t | subtropical monsoon climate | 41.00%→80.50% | 40.77%→52.01% | cropland→other |
| Anhui Province | 1990–2020 | ↓1.39 × 10$^7$ t | subtropical and warm temperate transition zone zone | 21.23%→58.33% | 24.03%→28.65% | cropland→building land |
| Jiangsu Province | 2000–2020 | ↓1.63 × 10$^8$ t | temperate monsoon climate and subtropical monsoon climate | 42.60%→73.44% | 7.54%→15.20% | cropland and forest→building land |
| Hebei Province | 1990–2015 | ↓4.44 × 10$^7$ t | temperate monsoon climate | 19.68%→51.67% | 13.12%→26.78% | cropland→other |

Note: ↑ represents an increase in data. ↓ represents a reduction in data. → He represents data changes from left to right.

In contrast, unutilized land in Shandong Province accounts for only 10.93%, while the land utilization rate in Gansu Province is only 65.07%, resulting in an increase in carbon stocks because of the transition from low-carbon-intensity unutilized land development to high-carbon-intensity cropland (1271 km$^2$) forest (27 km$^2$) and grassland (519 km$^2$) land. Meanwhile, grassland was largely converted to forest land, which has a higher carbon density (transfer area of 460 km$^2$, carbon density from 10.70 kg/m$^2$ to 18.89 kg/m$^2$). This conversion increased forest cover, highlighting the role of forest expansion in carbon sequestration. Although the expansion of building land encroached on part of the ecological land in Fujian Province and Kunming City (unlike in Shandong Province), the region's main land type is woodland, with a high forest cover and subtropical climate with a humid climate and high precipitation which is suitable for the growth of vegetation. Therefore, the carbon stock, although showing a declining trend, is relatively small.

In conclusion, the encroachment of construction land into original ecological areas is a primary driver of carbon stock reductions. Prospective studies ought to focus on the rational development of unutilized land, concentrating on approaches to augment the carbon sequestration capacity of regions with lower carbon densities and protect existing ecological areas to mitigate the impacts of land cover transformations on carbon stocks.

*4.2. Analysis of Regional Carbon Sink Potential in Shandong Province*

In Shandong Province, cities exhibit varying levels of carbon sink potential, driven by their carbon stock base and land use changes. Among cities with high carbon stocks, Yantai and Weifang demonstrate high carbon sink potential, while Linyi also shows considerable carbon sequestration capacity. Yantai City has a larger proportion of areas with higher carbon density, such as cultivated land, forest, and grassland, which make up 78.1% and 71.8% of the land area, respectively. From 2020 to 2030, carbon sequestration in these areas is projected to increase, with a rise of 11,092.05 tons, 101.79 tons, and 678.69 tons, respectively [51]. This reflects Yantai's strong carbon sink potential, driven by its large carbon-dense areas. Linyi City, despite having a significant carbon stock, has seen a decline in habitat quality. From 2000 to 2020, a habitat quality decrease of 0.053 led to a reduction of 5.39% in the carbon stock [47]. Moreover, this carbon stock is projected to continue declining under the Ecological Variation Circumstances (EVC), with a projected decrease of 8.88% from 2020 to 2040.

Among cities with larger carbon stocks, Qingdao, Jinan, and Dezhou demonstrate high carbon sink potential. Specifically, In Jinan City, the southern mountainous area saw an increase of $3.93 \times 104$ t in total carbon sequestration from 2010 to 2020 due to forest plantation optimization and an increase in forest cover [52]. Under the Ecological Variation Circumstances (EVC), Jinan's carbon reserves are anticipated to increase at a pace of 7.43% from 2020 to 2040. Heze City is facing a significant conflict between cultivated area and building area. From 2001 to 2020, plow land decreased by 1288.51 km$^2$, while construction land expanded by 1271.70 km$^2$ [53]. This has resulted in land degradation, as evidenced by a 3.53 km$^2$ increase in bare land. The carbon stock in Heze is expected to decline by $45.74 \times 10^6$ t from 2020 to 2040, representing a decrease of 24.75%.

Among the cities with general carbon stocks, Rizhao City exhibited a net increase in carbon sequestration, with a rise of $3.90 \times 10^4$ t from 1995 to 2015. However, the less carbon-dense waters along the eastern coast and unused land in Rizhao remain underutilized. Under EVC, the rate of carbon stock decline in Rizhao has decreased from 13.36% to 4.89%, showing a more stable carbon sequestration trend [54].

## 5. Conclusions

This study simulates and predicts the chron-ospatial variations in carbon stocks in Shandong Province from 1990 to 2040, utilizing land use data and other foundational data, as well as employing the CA-Markov and InVEST models. It investigates the correlation between land utilization changes and carbon stocks and analyzes the geographical carbon sink potential. The key conclusions are as follows:

(1)    Spatial distribution and trends of carbon reserves in Shandong Province

The spatial pattern of carbon stocks in Shandong Province closely follows land use patterns, focused mainly in the central and eastern zones, with lower concentrations in other areas. Over the last three decades, a general decline in carbon reserves has been observed. The contribution of each land utilization category to this change, in descending order, is as follows: cultivated land, forest, grassland, construction land, water, and unused land. The rapid expansion of construction areas has significantly reduced the proportion of cultivated land and grassland, leading to a transition from the high carbon density land utilization

category to low carbon density alternatives. This is the primary driver of the observed decline in carbon stocks. Additionally, forest area increases have contributed positively to carbon sequestration, indicating the potential for land use management strategies that prioritize reforestation.

(2)    Spatial Differentiation of Carbon Stocks in Shandong Province

Shandong Province exhibits significant spatial differentiation in its carbon stocks, which can be categorized into three distinct groups of cities: cities with high carbon stocks, i.e., Linyi, Weifang, and Yantai, which show consistently high levels of carbon stock; Cities with large carbon stocks, i.e., Jinan, Jining, Qingdao, Dezhou, Binzhou, Liaocheng, Taian, Zibo, and Dongying, which have relatively larger but more variable carbon stocks; and cities with general carbon stocks, i.e., Weihai, Rizhao, and Zaozhuang, which have more moderate carbon stocks compared to the cities in the other two categories.

The three projection scenarios reveal distinct trends. Under Ecological Variation Conditions, carbon stocks show a significant increase, particularly along the eastern coast of central Shandong Province. This suggests that areas with more favorable ecological environments (e.g., forest areas) have the potential to absorb and sequester more carbon. In contrast, under City's Variation Conditions, carbon stocks are expected to decline significantly, with areas of reduction expanding outward from areas with high levels of construction land. These findings underscore the pivotal role of land utilization planning and ecological conservation in mitigating carbon loss and enhancing sequestration.

(3)    Carbon Sink Potential in Shandong Province

The carbon sink potential in Shandong Province varies significantly across its cities. Under the Ecological Variation Conditions, carbon sink potential is categorized as follows: cities with high carbon sink potential, i.e., Yantai, Dezhou, Weifang, Qingdao, and Jinan; cities with larger carbon sink potential, i.e., Binzhou, Weihai, Zibo, Liaocheng, Dongying, Linyi, Tai'an, Rizhao, and Zaozhuang; and cities with general carbon sink potential, i.e., Jining and Heze.

The cities with high carbon sink potential are experiencing dynamic increases in carbon sequestration capacity. Local governments are prioritizing the preservation and expansion of forested areas and investing in ecological restoration initiatives. As such, cities can enhance their carbon storage capabilities while simultaneously improving urban green spaces. On the other hand, cities with larger carbon sink potential have reached a stabilization point in their carbon sequestration capacity, suggesting that while they are contributing positively to carbon storage, targeted efforts such as the implementation of sustainable land management practices (e.g., conservation agriculture) may be required to boost their carbon storage rates. In contrast, cities with general carbon sink potential are experiencing a decline in carbon stocks, which highlights the challenges posed by urbanization and land use change. Policymakers in these areas should pay attention to strategies to mitigate the influences of urban sprawl, such as promoting compact urban design, enhancing public transportation systems, and encouraging the development of green infrastructure.

**Author Contributions:** Conceptualization, writing—original draft preparation, X.X.; methodology, data, K.L.; investigation, supervision, J.Z. and Y.L.; project administration, and funding acquisition, C.L. and F.H. All authors have read and agreed to the published version of the manuscript.

**Funding:** This research was funded by Monitoring and Evaluation of Carbon Sinks in Natural Ecosystems of Shandong Province (Research and Construction of Carbon Measurement Model for Forest Land in Shandong Province) 2024; Natural Science Foundation of Shandong Province (grant number: ZR2021MD080); Program for Scientific Research Innovation Team of Young Scholar

in Colleges and Universities of Shandong Province (grant number: 2024KJG043). Biodiversity Conservation and Utilization in Mount Tai (grant number: 2022TSGS002).

**Data Availability Statement:** Data is contained within the article.

**Conflicts of Interest:** The authors declare no conflicts of interest.

## Abbreviations

The following abbreviations are used in this manuscript:

NVC   Natural Variation Conditions
EVC   Ecological Variation Conditions
CVC   City's Variation Conditions

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
