# Peer review of "Spatial and Chronological Assessment of Variations in Carbon Stocks in Land-Based Ecosystems in Shandong Province and Prospective Predictions (1990 to 2040)"

_sustainability, doi:10.3390/su17062424_

Round 1
Reviewer 1 Report
Comments and Suggestions for Authors
This study, which focuses on Shandong Province from 1990 to 2020, combines the CA-Markov and InVEST models to analyze temporal and spatial variations in carbon stocks while also projecting future changes under three different scenarios; the work emphasizes urbanization and land use transitions as major drivers of carbon stock decline, while ecological conservation policies are suggested to mitigate these impacts. The findings provide important insights into the field of land-use management and contribute valuable data to the ongoing discourse surrounding carbon neutrality strategies, however, the study lacks clarity in its objectives, rigor in methodology, and depth in its discussions.
Despite the potential significance of the study, there are several issues that need to be addressed for improving the manuscript’s overall quality, especially in areas such as result interpretation and policy recommendations.
1. Issues in the Introduction
(1)Repetition in Research Background (Lines 43–51): The background, which aims to set the context, reiterates the importance of carbon stocks multiple times, especially in lines 43–45, which are again somewhat echoed in 50–51 lines; this repetition dilutes the focus and makes the introduction feel redundant in some parts.
(2)Ambiguous Research Objectives (Lines 72–104): There is no specific or clearly delineated research hypothesis; the stated objective appears vague because it merely mentions exploring “carbon stock changes,” but fails to articulate the core questions driving the study. It need more clear definition.
(3)Insufficient Theoretical Basis for Methodology (Lines 53–71): The paper mentions CA-Markov and InVEST as advantageous, but the reasons behind their applicability in the specific context of Shandong Province remain somewhat unconvincing as it lacks sufficient theoretical underpinning to explain these choices comprehensively.
(4)Outdated Literature References (References [12–14]): References cited for the methodology are mostly outdated; it dosen't include new or relevant advancements post-2020, which are critical in fast-evolving research areas.
2. Issues in Materials and Methods
(1)Data Preprocessing Details (Lines 112–127): The data conversion processes, specifically how temperature and precipitation were transformed into raster formats, lacks detailed explanations regarding the interpolation methods used; error handling procedures are completely omitted, which leaves uncertainties about the data's reliability.
(2)Resolution Issues (Lines 113–121): The manuscript fails to provide details on how varying data resolutions were reconciled; while land-use data has 30m resolution, socio-economic data is at 1km, which makes one wonder about the potential inaccuracies introduced due to such mismatches.
(3)Simplistic Model Validation (Lines 191–195): The reliance on Kappa coefficients alone, with no details about sample selection, makes the validation seem rather incomplete and raises questions about its generalizability.
(4)Scenario Assumption Weaknesses (Lines 199–214): The assumptions underlying the City’s Variation Conditions (CVC) scenario, particularly that all land except water will be converted to urban use, appear oversimplified and lack realistic policy or economic contexts for justification.
3. Issues in Results
(1)Insufficient Data Interpretation (Lines 253–301): While results show temporal and spatial trends, such as Weifang's sharp decline in carbon stocks, the lack of an explanation for these observations leaves the readers with unanswered questions about the underlying drivers. It just don't explore the details.
(2)Dynamic Change Rates (Lines 253–267): The analysis of change rates in carbon stocks lacks depth; the results are presented in isolation, disconnected from climatic or socio-economic contexts, which diminishes the findings’ overall impact.
4. Issues in Discussion
(1)Simplistic Mechanistic Analysis (Lines 380–417): While acknowledging land-use changes as drivers of carbon stock variations, the paper does not delve into the mechanisms, such as how different land-use transitions affect soil organic carbon or aboveground biomass carbon.
(2)Superficial Regional Comparisons (Lines 389–417): When comparing Shandong to Anhui, the discussion remains surface-level; it does not analyze why Shandong's trends differ or how its unique characteristics influence its carbon stock changes.
(3)Non-Specific Policy Implications (Lines 417–419): The policy suggestions are generic and lack actionable detail, failing to provide a clear roadmap for ecological restoration or urban planning adjustments.
5. Issues in Conclusion
(1)Repetition (Lines 458–489): The conclusion echoes content already presented in the results section, providing little additional insight or abstraction.
(2)Unclear Practical Relevance (Lines 491–507): The application of findings to urban planning and agriculture is mentioned but not elaborated upon; there is no discussion of how these results could be implemented in real-world settings.
(3)Weak Objective Linkage (Lines 491–507): The conclusion does not systematically revisit the research objectives outlined in the introduction, leaving gaps in the logical flow of the manuscript.
6. Issues in References
(1)Outdated Citations (Lines 523–611): The references are heavily skewed toward studies published before 2020; this neglects recent research, which is especially critical in a rapidly advancing field like carbon stock estimation.
(2)Omissions of Key Literature: Foundational studies on CA-Markov and InVEST model integrations are missing, which weakens the methodological justification.
Author Response
Dear Editor:
Thank you for your letter and for the reviewers’ comments concerning our manuscript entitled “Spatial and Chronological Assessment of Variations in Carbon Stocks in Land-based Ecosystems in Shandong Province and Prospective Predictions (1990 to 2040)” (Manuscript ID: sustainability-3421139). All comments are extremely valuable and helpful to us for revising and improving our paper. We have carefully studied these comments and have made the redacted changes in the original text, which we hope you will recognize.
Comments 1: Repetition in Research Background (Lines 43–51): The background, which aims to set the context, reiterates the importance of carbon stocks multiple times, especially in lines 43–45, which are again somewhat echoed in 50–51 lines; this repetition dilutes the focus and makes the introduction feel redundant in some parts.
Response 1: Thank you for your valuable comments on my thesis. We have revised the relevant content in lines 45-53, streamlined and merged the repetitive parts in lines 45-46 to avoid redundancy, and highlighted the core of the study in lines 49-53. Accurately assessing the impacts of land-use change on regional carbon stocks is essential for optimizing land resource management and achieving the dual-carbon goal. A comprehensive analysis of the linkages between carbon stocks and land-use change could provide valuable insights for sustainable development.
Comments 2: Ambiguous Research Objectives (Lines 72–104): There is no specific or clearly delineated research hypothesis; the stated objective appears vague because it merely mentions exploring “carbon stock changes,” but fails to articulate the core questions driving the study. It needs a clear definition.
Response 2: Thank you for your detailed review and constructive comments on my thesis. We have revised the following content in lines 88-97. The aim of this study is capitalizing on land utilization data collected be-tween 1990 and 2020, combining CA-Markov and InVEST model to analyze the spatial and temporal variability of carbon stocks in Shandong Province and the response rela-tionship between land use changes and carbon stocks, to explore the differences in the impact of land use types on carbon stocks among regions, and to summarize the char-acteristics of the impact of land use types on carbon stocks in the study area. The study also seeks to forecast and model land use patterns and changes in carbon stocks for the years 2030 and 2040 across different scenarios, thus providing a detailed analysis of carbon stock variations. This research is expected to offer important insights for eco-logical development and dual-carbon strategy planning in Shandong Province.
Comments 3: Insufficient Theoretical Basis for Methodology (Lines 53–71): The paper mentions CA-Markov and InVEST as advantageous, but the reasons behind their applicability in the specific context of Shandong Province remain somewhat unconvincing as it lacks sufficient theoretical underpinning to explain these choices comprehensively.
Response 3:Thank you for your detailed review and valuable comments on my paper. I have added detailed explanations of the theoretical background of the CA-Markov and InVEST models in lines 59-62 and 65-68, especially their applicability in the specific context of Shandong Province. I point out how these models effectively address the limitations of traditional approaches and cite relevant literature on their successful application in similar studies [10-11].
Comments 4: Outdated Literature References (References [12–14]): References cited for the methodology are mostly outdated; it doesn't include new or relevant advancements post-2020, which are critical in fast-evolving research areas.
Response 4: We thank the reviewers for their careful review and valuable comments on our paper. We reviewed the latest literature and added relevant research results published after 2020. This literature reflects the latest advances in the field and enhances the timeliness and cutting edge of our methodology section. The following recent studies were cited:
12.Paruka, W.J.; Ai, D.; Fang, Y.S.; Zhang, Y.B.; Li,M.; Hao, J.M. Spatial and Temporal Evolution and Prediction of Carbon Storage in Kunming City Based on InVEST and CA⁃Markov Model. Environmental Science.2024, 45 (01), 287-299.
13.Yang, J.; Xie, B.P.; Zhang, D.G. Spatio-temporal evolution of carbon stocks in the Yellow River Basin based on InVEST and CA-Markov models. Chinese Journal of Eco-Agriculture. 2021, 29(6), 1018-1029.
14.Wu, L.Q.; Chen, R.Q. Spatial and temporal evolution and prediction of ecosystem carbon storage combining InVEST and CA-Markov model: a case study of Baoji, Shanxi Province. Shanghai Land & Resources. 2024, 45(4), 79-86.
Comments 5: Data Preprocessing Details (Lines 112–127): The data conversion processes, specifically how temperature and precipitation were transformed into raster formats, lacks detailed explanations regarding the interpolation methods used; error handling procedures are completely omitted, which leaves uncertainties about the data's reliability.
Response 5: We thank the reviewers for their careful review of our paper and helpful suggestions. We neglected to describe the details of the data processing when writing the manuscript and have added a detailed description of the interpolation method in the data preprocessing section (see lines 120-124, 128-130). Specifically, we rasterize the temperature and precipitation data using kriging interpolation. We have added a detailed description of the error handling process (see lines 120-124). A detailed explanation of how missing values and outliers in the data are handled to ensure data reliability is provided.
Comments 6: Resolution Issues (Lines 113–121): The manuscript fails to provide details on how varying data resolutions were reconciled; while land-use data has 30m resolution, socio-economic data is at 1km, which makes one wonder about the potential inaccuracies introduced due to such mismatches.
Response 6: We thank the reviewers for their careful review of our manuscript and helpful suggestions. We neglected to describe the resolution harmonization process when writing the manuscript, which affected the readers. We have added to the manuscript a detailed description of how we handled the data at different resolutions, using a resampling technique that aggregates 30 m land use data to 1 km resolution. The rationale for choosing this method is explained and its impact on the spatial consistency of the data is discussed (see lines 137-138).
Comments 7: Simplistic Model Validation (Lines 191–195): The reliance on Kappa coefficients alone, with no details about sample selection, makes the validation seem rather incomplete and raises questions about its generalizability.
Response 7: We thank the reviewers for their careful review of our manuscript and constructive suggestions. We chose a smaller sample size for model validation, which has an impact on the accuracy of the data. We have added a detailed description of the sample selection process, including the source and number of samples, to the manuscript (see lines 202-207), which helps to improve the transparency and generalizability of the validation.
Comments 8: Scenario Assumption Weaknesses (Lines 199–214): The assumptions underlying the City’s Variation Conditions (CVC) scenario, particularly that all land except water will be converted to urban use, appear oversimplified and lack realistic policy or economic contexts for justification.
Response 8: Thank you for your careful review of our manuscript and valuable suggestions. Our assumption that all land except watersheds will be converted to urban land is intended to explore the urbanization process under extreme scenarios and is expressed in the article (lines 220-222). We realize that this assumption may be too simplistic, especially in the absence of a realistic policy or economic context to support it. Therefore, we have added a discussion of current urbanization policies and forms of ecological conservation to the revised version, citing relevant literature and data to support this context (lines 209-211).
Comments 9: Insufficient Data Interpretation (Lines 253–301): While results show temporal and spatial trends, such as Weifang's sharp decline in carbon stocks, the lack of an explanation for these observations leaves the readers with unanswered questions about the underlying drivers. It just doesn't explore the details.
Response 9: Thank you for your valuable comments on our manuscript. We have added discussions on land use change, climatic factors and socio-economic factors. In particular, regarding the reduction of arable land, the expansion of building land and climate change, we further analyzed the impacts of these factors on carbon stocks in the light of existing literature and data: first, the expansion of building land led to a direct loss of carbon stocks (-15810.23t) due to the urbanization of carbon-rich ecosystems such as forests and grasslands. Secondly, the decrease in the area of cultivated land per unit area, mainly caused by urbanization and farmland abandonment, led to a loss of carbon stock of 9,896.51t. In addition, the increase in forest carbon stock (17585.78t) during this period, although significant, was not enough to offset the loss of other land types. The overall decline also reflects broader land use change and may be influenced by regional climate patterns such as reduced precipitation and temperature fluctuations, which can directly affect soil carbon storage and vegetation growth (rows 268-278). Discussions of land use change, urbanization and economic factors were added to explain the reasons for the changes in carbon stocks in different cities: 16 cities in Shandong Province showed a clear downward trend in carbon stocks throughout the study period. In Weifang City, the largest decrease in carbon stock was -33.27×106 t. The reason for the significant decrease in carbon stock was the accelerated urbanization process, which transformed agricultural land into urban land, resulting in a large amount of carbon loss. In contrast, Zaozhuang City had the smallest decrease of -7.74×106 t. This indicates that the more stable land use pattern and effective land management measures in Zaozhuang City helped to maintain its carbon stock. This shift may be influenced by land use changes such as increased agricultural activities or urban expansion, which can negatively affect carbon sequestration. Similarly, Zibo transitioned from the larger carbon stock category (110.43 ×106t) to the average carbon stock category (99.93×106t), suggesting a gradual degradation of its carbon stock, possibly due to industrial activities and urban development (lines 303-316).
Comments 10: Dynamic Change Rates (Lines 253–267): The analysis of change rates in carbon stocks lacks depth; the results are presented in isolation, disconnected from climatic or socio-economic contexts, which diminishes the findings’ overall impact.
Response 10: Thank you for your valuable comments on our manuscript. We have added a discussion of the reasons behind the rate of change in carbon stocks, especially climate change, urbanization, land use, and industrialization. We have explored how socio-economic and climatic factors affect the trend of carbon stock changes in the context of carbon stock changes in different cities.The contribution of cities with high carbon stock to the overall provincial average accounted for 38.61%, with Linyi City standing out due to its impressive average an-nual carbon stock of 289.16×106 t, despite a decrease of 33.12×106 t over the 30-year pe-riod. This decrease could be ascribed to both natural factors, for instance climate vari-ability and human factors, including agricultural land conversion and industrial ex-pansion. Linyi's decrease rate (k = -3.16) indicates a relatively moderate decline com-pared to other cities. Weifang City, exhibiting the highest decrease rate (k = -11.25), experienced the most significant reduction in carbon stock, totaling 33.27×106 t. This dramatic decline can be attributed to rapid urbanization and the conversion of car-bon-rich land into urban infrastructure. Additionally, Weifang has been undergoing significant industrial development, which may have led to the degradation of local carbon sinks. In contrast, cities categorized with larger carbon stocks contributed 43.11% to the provincial average, showing that they managed to maintain relatively stable carbon reserves despite slight decreases. On the other hand, cities with general carbon stocks showed relatively smaller decreases in carbon stock (k values ranging from -5.83 to -2.65), indicating that changes in their carbon stock were comparatively stable. These cities tend to have a more balanced approach to land use and urbanization, which may have helped preserve their carbon reserves. The more stable trends in these cities could be reflective of less rapid industrialization, better land management prac-tices, and a more stable climate compared to other regions in Shandong Province.(rows 318-337).
Comments 11: Simplistic Mechanistic Analysis (Lines 380–417): While acknowledging land-use changes as drivers of carbon stock variations, the paper does not delve into the mechanisms, such as how different land-use transitions affect soil organic carbon or aboveground biomass carbon.
Response 11: Thank you for reviewing our paper and for your valuable suggestions. We have added a mechanism for how land use change affects carbon stocks by influencing soil organic carbon and aboveground biomass carbon in lines 424-441. The increase in built-up land usually involves the conversion of natural landscapes into urban or industrial areas, which results in a net loss of carbon. This is mainly due to the loss of vegetation and soil carbon stocks as a result of the metamorphosis of built-up land. In addition, erosion of the soil surface and reduction of vegetation severely limit the potential for carbon sequestration in these areas. On the other hand, the increase in forest area contributes to higher carbon sequestration due to the increase in vegetation biomass in forest ecosystems and the increase in soil organic carbon accumulation. Forest soils are rich in organic matter and generally store more carbon than other land use types because the slower rate of decomposition in forest environments contributes to the long-term preservation of organic carbon. The reduction of cropland results in a net increase in carbon stocks due to the cessation of soil degradation and the restoration of soil organic carbon through reduced tillage. The shift from cropland to other land uses has allowed for the restoration of soil carbon sequestration, which is critical to reducing carbon emissions. In addition, the reduction in the area of grassland has led to a slight increase in carbon stocks, as grasslands are typically carbon sinks due to their deep root systems, but may lose carbon through degradation or overgrazing.
Comments 12: Superficial Regional Comparisons (Lines 389–417): When comparing Shandong to Anhui, the discussion remains surface-level; it does not analyze why Shandong's trends differ or how its unique characteristics influence its carbon stock changes.
Response 12: Thank you for your detailed review and valuable comments on our paper. We have analyzed the impacts of climatic conditions, land-use patterns and urban changes in Shandong on its carbon stock changes in rows 442-476. Tabular data have been added to make the comparative analysis between regions more direct and clear. The conversion of other types of land to building land is the main reason for the decrease in carbon stock, while the development and utilization of unutilized land is the main reason for the increase in carbon stock. These regional differences emphasize that carbon intensity is the main factor influencing changes in carbon stocks. Due to differences in geographic, natural and climatic conditions, carbon density in Northwest China is generally lower than that in East China. Specifically, differences in precipitation and temperature between Gansu and Inner Mongolia Autonomous Region and Shandong and Anhui result in different carbon density correction factors. However, regional differences are not only caused by climatic factors; differences in urbanization also have a significant impact on land use conversion and carbon density. In Shandong province, the urbanization level increased from 27.30% to 63.05% in 30 years, and the reduced arable land was mainly converted to construction land (transferring an area of 17,895.2 km2 , and the carbon density decreased from 190.49 t/km2 to 24.29 t/km2 ), which led to the reduction of carbon stock. The main reasons for the decrease in carbon stock in Jiangsu, Hebei, Anhui and Hangzhou provinces are consistent with those of Shandong province, which may be related to the fact that these areas have the same land use change pattern as Shandong province, less climatic differences, and accelerated urbanization. In contrast, the unutilized land in Shandong province accounts for only 10.93%, while the land use rate in Gansu province is only 65.07%, leading to an increase in carbon stock due to the transition from the de-exploitation of the unutilized land with low carbon intensity to the cropland (1,271km2 ), woodland (27km2 ), and grassland (519km2 ) with high carbon intensity. Most of the reduced grassland was converted to forest land with higher carbon intensity (the converted area was 460 km2 and the carbon intensity increased from 10.70 kg/m2 to 18.89 kg/m2). This conversion increased forest cover, highlighting the role of forest expansion in carbon sequestration. Unlike Shandong Province, Fujian Province and Kunming City, although the expansion of construction land encroached on some ecological land, the region's main land type is forested, with high forest cover and a subtropical climate with a humid climate and high precipitation, which is suitable for vegetation growth. As a result, carbon stocks, although on a downward trend, are relatively small.
|
Region |
Year |
Carbon stock changes |
Climatic region |
Urbanization level |
Forest cover |
Main causes of carbon stock changes |
|
Hangzhou City |
2000-2020 |
↓2.41×106t |
temperate monsoon climate |
61.17%→83.30% |
54.41%→59.43% |
cropland→building land, forest→other |
|
Gansu Province |
1990-2015 |
↑2.51×106t |
temperate continental climate |
23.54%→43.19% |
6.66%→11.33% |
unused land→other |
|
Inner Mongolia Autonomous Region |
2000-2020 |
↓1.01×108t |
temperate continental climate |
42.68%→67.48% |
17.7%→22.10% |
cropland→building land |
|
Fujian Province |
2000-2020 |
↓4.47×106t |
subtropical monsoon climate |
41.99%→68.75% |
62.96%→66.80% |
cropland and grassland→ building land |
|
Kunming City |
2000-2020 |
↓9.85×105t |
subtropical monsoon climate |
41.00%→80.50% |
40.77%→52.01% |
cropland→other |
|
Anhui Province |
1990-2020 |
↓1.39×107t |
subtropical and warm temperate transition zone zone |
21.23%→58.33% |
24.03%→28.65% |
cropland→building land |
|
Jiangsu Province |
2000-2020 |
↓1.63×108t |
temperate monsoon climate and subtropical monsoon climate |
42.60%→73.44% |
7.54%→15.20% |
cropland and forest→building land |
|
Hebei Province |
1990-2015 |
↓4. 44×107t |
temperate monsoon climate |
19.68%→51.67% |
13.12%→26.78% |
cropland→other |
Comments 13: Non-Specific Policy Implications (Lines 417–419): The policy suggestions are generic and lack actionable detail, failing to provide a clear roadmap for ecological restoration or urban planning adjustments.
Response 13: Thank you for your valuable suggestions on this paper. We adjusted the focus of the section to the response relationship between carbon stock and land use change and the analysis of carbon sink potential, and deleted the section on policy recommendations to avoid the discussion being too general. Therefore, policy recommendations on ecological restoration or urban planning adjustment will be further explored in other suitable studies.
Comments 14: Repetition (Lines 458–489): The conclusion echoes content already presented in the results section, providing little additional insight or abstraction.
Response 14: Thank you for your insights into the concluding section of our manuscript. We have removed redundant numerical data in favor of a more abstract discussion of the findings and their implications. (1) Spatial distribution and trends of carbon reserve in Shandong Province
The spatial pattern of carbon stock in Shandong Province closely follows land use patterns, focused mainly in the central and eastern zones, and lower concentrations in other areas. Over the last three decades, a general decline in carbon reserves has been observed. The contribution of each land utilization category to this change, in descending order, is as follows: cultivated land, forest, grassland, construction land, water and unused land. The rapid expansion of construction area has significantly reduced the field of cultivated land and grassland, leading to a transition from high carbon density land utilization category to low carbon density alternatives. This is the primary driver of the observed decline in carbon stock. Additionally, forest area increase has contributed positively to carbon sequestration, indicating the potential for land use management strategies that prioritize reforestation.
(2) Spatial Differentiation of Carbon Stocks in Shandong Province
Shandong Province exhibits significant spatial differentiation in its carbon stocks, which can be categorized into three distinct groups of cities. Cities with high carbon stocks: These include Linyi, Weifang, and Yantai, which show consistently high levels of carbon stock. Cities with large carbon stocks: This group comprises Jinan, Jining, Qingdao, Dezhou, Binzhou, Liaocheng, Taian, Zibo, and Dongying, which have relatively larger but more variable carbon stocks. Cities with general carbon stocks: These cities, including Weihai, Rizhao, and Zaozhuang, have more moderate carbon stock levels compared to the other two categories.
The three projection scenarios reveal distinct trends. Under Ecological Variation Conditions, carbon stocks show a significant increase, particularly along the eastern coast of central Shandong Province. This suggests that areas with a more favorable ecological environment (e.g., forest areas) have the potential to absorb and sequester more carbon. In contrast, under City's Variation Conditions, carbon stocks are expected to decline significantly, with areas of reduction expanding outward from areas with high levels of construction land. These findings underscore the pivotal role of land utilization planning and ecological conservation in mitigating carbon loss and enhancing sequestration(lines 515-543).
Comments 15: Unclear Practical Relevance (Lines 491–507): The application of findings to urban planning and agriculture is mentioned but not elaborated upon; there is no discussion of how these results could be implemented in real-world settings.
Response 15: Thank you for your constructive feedback on the clarity of the practical implications of our findings. We have highlighted specific strategies for implementing our findings, including recommendations for local governments and policymakers to enhance carbon sequestration through sustainable practices. The carbon sequestration capacity of cities with high carbon sink potential is increasing. With local governments prioritizing the protection and expansion of forest areas and investing in ecological restoration measures, these cities can improve urban green spaces while increasing their carbon storage capacity. On the other hand, the carbon sequestration capacity of cities with high carbon sink potential has reached a stabilization point, suggesting that while these cities are contributing positively to carbon sequestration, targeted efforts, such as the implementation of sustainable land management measures (e.g., conservation agriculture), are still needed to improve carbon sequestration rates. In contrast, carbon stocks in cities with average carbon sink potential are declining, highlighting the challenges posed by urbanization and land use change. Policymakers in these regions should focus on strategies to mitigate the impacts of urban sprawl, such as promoting compact urban design, strengthening public transportation systems, and encouraging the development of green infrastructure (lines 544-563).
Comments 16: Weak Objective Linkage (Lines 491–507): The conclusion does not systematically revisit the research objectives outlined in the introduction, leaving gaps in the logical flow of the manuscript.
Response 16: Thank you for your insightful comments regarding the link between the conclusions and the research objectives outlined in the introduction. We have revised the conclusions to systematically revisit the research objectives and summarize how our findings align with each objective. In this study, spatial and temporal changes in carbon stocks in Shandong Province from 1990 to 2040 were simulated and projected using land-use data and other basic data with CA-Markov and InVEST models. The correlation between land use change and carbon stock was studied, and the potential of geographical carbon sinks was analyzed. The main conclusions are as follows (lines 510-514).
Comments 17: Outdated Citations (Lines 523–611): The references are heavily skewed toward studies published before 2020; this neglects recent research, which is especially critical in a rapidly advancing field like carbon stock estimation.
Response 17: Thank you for your valuable comments on the references section of this paper. We have updated the references to ensure that more relevant studies published in recent years are cited, especially the latest results in the field of carbon stock estimation. We have reviewed the relevant literature in recent years and incorporated the latest research results, which are updated and redlined in the article and references.
Comments 18: Omissions of Key Literature: Foundational studies on CA-Markov and InVEST model integrations are missing, which weakens the methodological justification.
Response 18: Thank you for your valuable comments on our methodology section. We have reviewed and cited relevant literature to complement the theoretical background and practical applications of the integration of these two models, especially in the context of land use change and carbon stock estimation. Studies such as references [12-16] demonstrate the effectiveness of using CA-Markov models in conjunction with the InVEST framework.
Special thanks to you for your helpful comments.
We have endeavored to improve our manuscript and have made the required changes in the manuscript. These changes do not influence the conclusion and framework of the paper.
We appreciate Editors and reviewers’ helpful suggestions and comments and hope that our corrections meet with approval.
Once again, we thank you very much for your comments and suggestions.
Reviewer 2 Report
Comments and Suggestions for Authors
Manuscript Number: sustainability-3421139
Title:Spatial and Chronological Assessment of Variations in Carbon Stocks in Land-based Ecosystems in Shandong Province and Prospective Predictions (1990 to 2040)
I reviewed the manuscript titled "Spatial and Chronological Assessment of Variations in Carbon Stocks in Land-based Ecosystems in Shandong Province and Prospective Predictions (1990 to 2040)" This study provides valuable insights into the spatial and temporal dynamics of carbon stocks in Shandong’s land-based ecosystems, particularly in relation to land use changes. The research methodology is sound, combining spatial assessments with future projections, making it highly relevant for understanding regional carbon dynamics under climate change. However, the manuscript requires several revisions before it can be considered for publication, particularly in terms of clarifying the data sources, detailing the modeling approach, and expanding the discussion to better compare the findings with existing studies and acknowledge the study's limitations.
Abstract: The importance of regional carbon stocks is not clearly explained, nor is the rationale for conducting this research in Shandong emphasized. The significance of Shandong should be highlighted, and the potential value of future predictions should be better articulated to explain what benefits the future predictions will bring.
Introduction: Although the introduction provides background on the research, it lacks an in-depth discussion of other studies in related fields. A more detailed review of both domestic and international research on carbon stock variations and land use changes in similar regions would further highlight the novelty of this study.
Research Hypotheses: The research hypotheses or expected outcomes should be clearly listed. For example, is it expected that carbon stocks will follow a specific trend as land use changes? This would greatly help guide the reader in understanding the logic of the research.
Materials and Methods:
Clear Description of Models and Methods: The models and analysis methods used should be described in greater detail. For instance, did the study use specific spatial analysis models (such as GIS models or statistical regression analysis)? If so, were they effective in capturing the relationship between carbon stocks and land use? Additionally, the process of model validation and the selection of parameters should also be explained.
Results Analysis:
Group Analysis: Was a group analysis performed for different regions and land use types? If so, the carbon stock variations for each region or land use type should be discussed in detail to enhance the depth of the analysis.
Correlation with Land Use Changes: Is the variation in carbon stocks closely linked to changes in land use (such as agricultural expansion or deforestation)?
Consideration of External Factors: Were external factors such as climate change considered in relation to the impact on carbon stocks?
Accuracy and Reliability of Prediction Models: Are the prediction models accurate, and are their results reliable?
Author Response
Dear Editor:
Thank you for your letter and for the reviewers’ comments concerning our manuscript entitled “Spatial and Chronological Assessment of Variations in Carbon Stocks in Land-based Ecosystems in Shandong Province and Prospective Predictions (1990 to 2040)” (Manuscript ID: sustainability-3421139). All comments are extremely valuable and helpful to us for revising and improving our paper. We have carefully studied these comments and have made the redacted changes in the original text, which we hope you will recognize.
Comments 1: Abstract: The importance of regional carbon stocks is not clearly explained, nor is the rationale for conducting this research in Shandong emphasized. The significance of Shandong should be highlighted, and the potential value of future predictions should be better articulated to explain what benefits the future predictions will bring.
Response 1:Thanks to your valuable comments, we have clarified the importance of regional carbon stocks in addressing climate change (rows 15-17) and emphasized the background and need for this study in Shandong Province in the summary (rows 17-21). In addition, we describe the potential value of future projections and explain the benefits of the study for regional carbon management, policymaking, and sustainable development (rows 37-39).
Comments 2:Introduction: Although the introduction provides background on the research, it lacks an in-depth discussion of other studies in related fields. A more detailed review of both domestic and international research on carbon stock variations and land use changes in similar regions would further highlight the novelty of this study.
Response 2: We would like to thank the reviewers for their careful review and valuable comments on our paper. We have simplified the first paragraph of the introduction to make the presentation more concise and powerful (lines 45-53). The reasons for the applicability of the selected model in Shandong Province are explored in depth by citing several important papers [10-11] (lines 59-68), and several papers with recent research results [12-14] are cited to discuss the results of the research on carbon stock changes and land-use changes in similar regions.
Comments 3: Research Hypotheses: The research hypotheses or expected outcomes should be clearly listed. For example, is it expected that carbon stocks will follow a specific trend as land use changes? This would greatly help guide the reader in understanding the logic of the research.
Response 3:Thank you for your careful review of our manuscript and valuable suggestions. We have revised the following content in lines 76-97. The aim of this study is capitalizing on land utilization data collected be-tween 1990 and 2020, combining CA-Markov and InVEST model to analyze the spatial and temporal variability of carbon stocks in Shandong Province and the response rela-tionship between land use changes and carbon stocks, to explore the differences in the impact of land use types on carbon stocks among regions, and to summarize the char-acteristics of the impact of land use types on carbon stocks in the study area. The study also seeks to forecast and model land use patterns and changes in carbon stocks for the years 2030 and 2040 across different scenarios, thus providing a detailed analysis of carbon stock variations. This research is expected to offer important insights for eco-logical development and dual-carbon strategy planning in Shandong Province.Comments 4: Clear Description of Models and Methods: The models and analysis methods used should be described in greater detail. For instance, did the study use specific spatial analysis models (such as GIS models or statistical regression analysis)? If so, were they effective in capturing the relationship between carbon stocks and land use? Additionally, the process of model validation and the selection of parameters should also be explained.
Response 4: Thank you for your careful review of our manuscript and valuable suggestions. We describe in detail the specific model types used, including data processing in ArcGIS software and land use simulation in IDRISI 17.2 (lines 121-138), and explain their rationale and applicability in this study. Model validity: We discuss how the InVEST model and CA-Markov model effectively reflect the relationship between carbon stocks and land use (lines 140-145, 182-189) and explain the rationale for scenario simulation (lines 209-211), citing relevant literature to support this argument [13,44]. Model validation process and parameter selection: We add a detailed description of the model validation process and increase the number of validation samples (lines 202-207) to ensure the accuracy and reliability of the model.
Comments 5: Group Analysis: Was a group analysis performed for different regions and land use types? If so, the carbon stock variations for each region or land use type should be discussed in detail to enhance the depth of the analysis.
Response 5: Thank you for your review and suggestions on our manuscript. We provide a detailed subgroup analysis of different carbon stock classes in different regions, clarify the criteria for the analysis (lines 175-178), and provide an in-depth discussion of carbon stock changes in each subgroup (lines 318-337). We add a discussion of the causes of carbon stock changes in each region, including the effects of urbanization, agricultural management, and forest protection, and cite relevant literature (rows 318-337) to enhance the depth of the analysis.
Comments 6: Correlation with Land Use Changes: Is the variation in carbon stocks closely linked to changes in land use (such as agricultural expansion or deforestation)?
Response 6: Thank you for your review and valuable comments on our manuscript. Through numerical analysis, we quantified the relationship between land use change and carbon stock change in the study area, and discussed in detail the impacts of factors such as agricultural expansion, deforestation, and urbanization process on carbon stock, and clarified how these changes led to the reduction of carbon stock (lines 269-278).
Comments 7: Consideration of External Factors: Were external factors such as climate change considered in relation to the impact on carbon stocks?
Response 7: Thank you for reviewing our manuscript and for your valuable comments. We discuss the interactions between land use change and climate change in the analysis of carbon stock changes in different regions in the context of land use change and climate change, emphasizing the importance of considering external factors in the analysis of carbon stock changes (lines 304-337).
Comments 8: Accuracy and Reliability of Prediction Models: Are the prediction models accurate, and are their results reliable?
Response 8: Thank you for your review and valuable comments on our manuscript. We describe in detail the comparative validation using multi-raster data in the Methods section, where all validation metrics ensure model fitness and accuracy, as well as model generalization ability and stability (lines 202-207).
Special thanks to you for your helpful comments.
We have endeavored to improve our manuscript and have made the required changes in the manuscript. These changes do not influence the conclusion and framework of the paper.
We appreciate Editors and reviewers’ helpful suggestions and comments and hope that our corrections meet with approval.
Once again, we thank you very much for your comments and suggestions.
Round 2
Reviewer 1 Report
Comments and Suggestions for Authors
I sincerely thank the author for respecting my comments and revising the manuscript. Without a doubt, the quality of this version of the manuscript has greatly improved, and I recommend its immediate acceptance and publication.